

# Absence of Luttinger's theorem for fermions with power-law Green functions

**Kridsanaphong Limtragool[1], Zhidong Leong[1] and Philip W. Phillips[1]**

**1** Department of Physics and Institute for Condensed Matter Theory,
University of Illinois, 1110 W. Green Street, Urbana, IL 61801, U.S.A.

## Abstract

We investigate the validity of Luttinger's theorem (or Luttinger sum rule) in two scale-invariant fermionic models. We find that, in general, Luttinger's theorem does not hold in a system of fermions with power-law Green functions which do not necessarily preserve particle-hole symmetry. However, Ref. [1,2] showed that Luttinger liquids, another scale-invariant fermionic model, respect Luttinger's theorem. To understand the difference, we examine the spinless Luttinger liquid model. We find two properties which make the Luttinger sum rule valid in this model: particle-hole symmetry and $\mathrm{Im}\, G(\omega = 0, -\infty) = 0$. We conjecture that these two properties represent sufficient, but not necessary, conditions for the validity of the Luttinger sum rule in condensed matter systems.

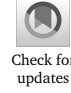

# 1  Introduction

A key problem in modern condensed matter physics involves identifying the propagating degrees of freedom in the normal state of cuprate superconductors. Since Landau's Fermi liquid theory fails to explain many features in the normal state, e.g., $T$-linear resistivity, the pseudogap, and Fermi arc formation, the low-energy degrees of freedom lie elsewhere. To progress further, one needs to know how the emergent charge carriers in the infrared are related to the bare electrons. For a Fermi liquid, Luttinger's theorem [3] relates the density of electrons at fixed chemical potential to the number of excitations in the Fermi liquid (i.e., Fermi surface volume) [4]. However, the original proof of the theorem for interacting electrons [3,5] relies on perturbation theory. This leads to the question of whether Luttinger's theorem still holds in a strongly correlated fermionic system such as the normal state of the cuprates. Equivalently, is there a version of this theorem that is valid independent of the Fermi liquid ansatz?

Mathematically, Luttinger's theorem for a system of spin-1/2 fermions states that the particle density $n$ is given in terms of the single-particle Green function $G(\mathbf{p}, \omega)$ by

$$n = 2 \sum_{\mathbf{p}} \theta(G(\mathbf{p}, \omega = 0)), \tag{1}$$

where $\theta(x)$ is the Heaviside function.[1] Recall that $G(\mathbf{p}, \omega \to -\infty) = \frac{1}{\omega} < 0$ for fermions, and notice that the Heaviside function is nonzero only when $G(\mathbf{p}, \omega = 0) > 0$. Consequently, only momenta at which $G(\mathbf{p}, \omega)$ changes sign from negative to positive as $\omega$ increases from $-\infty$ to 0 contribute to the sum. For a Fermi liquid, $G_{FL}(\mathbf{p}, \omega) = \frac{1}{\omega - \varepsilon_{\mathbf{p}}}$ with $\varepsilon_{\mathbf{p}}$ being the energy dispersion. Thus for a Fermi liquid, the summation counts the number of simple poles or the number of single particle excitations below the Fermi surface.

However, zeros also contribute to the sum in Eq. 1. Zeros are relevant to strongly correlated systems such as the cuprates in which one signature of the parent Mott insulator and the pseudogap phases is the appearance of zeros in the single-particle Green function [4, 6–10]. One of us [11] showed that, when a single-particle Green function has zeros, the Luttinger sum in Eq. 1 does not necessarily give the particle density. This stems from the fact that in the proof of Luttinger's theorem, the density has the form $n = I_1 + I_2$, with $I_1 = 2 \sum_{\mathbf{p}} \theta(G(\mathbf{p}, \omega = 0))$ and $I_2$ vanishing when the Luttinger-Ward (LW) functional exists. However, if the Green function has zeros (or, in other words, the self-energy diverges), the LW does not exist. Hence, $I_2$ is not guaranteed to be zero.

Another signature of the cuprates' normal state is the power law behavior of its physical properties. Since scale invariance and quantum criticality are widely used to explain these behaviors [12–17], it is important to study the validity of Luttinger's theorem for systems with scale-invariant Green functions. A concrete example would be the Green function of fermionic unparticles used in Ref. [18]. The Green function is of the form, $G \sim \frac{1}{(\omega - \varepsilon_{\mathbf{p}})^\alpha}$, where $\alpha$ is an anomalous exponent with $\alpha = 1$ corresponding to a Fermi liquid. While unparticles were originally proposed by Georgi [19] as a low-energy scale-invariant sector in the standard model, one of us [20] has applied the notion of unparticles to explain the breakdown of the particle picture in the cuprates. Models involving unparticles were later also used to explain the power laws in the transport properties [21] and the electronic scattering rate [18, 22] observed in the cuprates.

In this paper, we investigate the validity of Luttinger's theorem to spinless fermionic systems with a power-law Green function of the form, $G \sim \frac{1}{(\omega - \varepsilon_{\mathbf{p}})^\alpha}$. By explicitly calculating the density of fermions, we find that Luttinger's theorem does not hold in general. Only when

---

[1]We consider only spinless fermions in this paper. The spin degeneracy factor in Eq. 1 will be dropped in subsequent sections.

$1 < \alpha < 2$ with specific values of parameters can the Luttinger sum rule be satisfied. However, according to Ref. [1, 2], Luttinger's theorem is in fact valid for Luttinger liquids, another fermionic system with a scale-invariant Green function. To resolve this discrepancy, we directly verify Luttinger's theorem for the spinless Luttinger model [23] by explicitly computing the density. We identify two important properties necessary for Luttinger's theorem to be valid in this model: particle-hole symmetry and $\text{Im}G(\omega = 0, -\infty) = 0$. These properties are what is required for Luttinger sum rule to be valid in the Hubbard model [8] and the $SU(N)$ Hubbard model [11]. We conjecture that they are sufficient, but not necessary, conditions for the validity of Luttinger's theorem.

## 2 Fermions with power-law Green functions

We are interested in testing the validity of Luttinger's theorem when the fermionic Green function,

$$G(\lambda \varepsilon_{\mathbf{p}}, \lambda \omega) = \lambda^{-\alpha} G(\varepsilon_{\mathbf{p}}, \omega), \tag{2}$$

has a scaling form. Here, we specify that the Green function depends on momentum $\mathbf{p}$ through a dispersion relationship $\varepsilon_{\mathbf{p}}$. For concreteness, we consider the power-law Green function,

$$G(\mathbf{p}, \omega) = \frac{N}{(\omega - \varepsilon_{\mathbf{p}})^{\alpha}}, \tag{3}$$

where $\alpha$ is an anomalous exponent and $N$ is the normalization factor. The normalization factor $N$ can be specified by requiring that the spectral function $A \equiv -\frac{1}{\pi}\text{Im}G^{R}$ satisfy the sum rule,[2]

$$\int A(p, \omega)d\omega = 1, \tag{4}$$

with $G^{R}$ being the retarded Green function.

When $\alpha = 1$, this Green function simply describes quasiparticle excitations. Therefore, we focus on the case in which $\alpha$ is not an integer with $0 < \alpha < 2$. Hence, the Green function in Eq. 3 has a branch cut extending from $\omega = \varepsilon_{\mathbf{p}}$ in the complex $\omega$ space. We choose the branch cut to lie along the negative real axis with phase angle, $\phi$, defined in the range $-\pi < \phi \leq \pi$. As the Green function in Eq. 3 represents the low-energy theory of a system, by construction its range of validity is within an energy width $-E < \omega < E$, where $E$ is the UV or high energy cutoff, assumed to be much greater than $|\varepsilon_{\mathbf{p}}|$. We will see below in Eq. 8 that this assumption keeps the normalization factor momentum independent.

When $\alpha > 1$, the theory has an infrared divergence. So, it is necessary to impose a low energy cutoff, $\delta$, assumed to be much smaller than both $E$ and $\varepsilon_{\mathbf{p}}$. We explicitly include this $\delta$ in both the $0 < \alpha < 1$ and $1 < \alpha < 2$ cases. We treat $\delta$ as finite when $1 < \alpha < 2$ and set $\delta = 0$ when $0 < \alpha < 1$ at the end of the calculation. The infrared cutoff $\delta$ represents the breaking of scale invariance in a similar fashion to that in Ref. [24].

### 2.1 Luttinger's theorem for fermions with power-law Green functions

Luttinger's theorem in the form of Eq. 1 implicitly assumes the Green function at frequencies $\omega = -\infty$ and $\omega = 0$ to be real (or equivalently the imaginary part of the self-energy is zero at these two frequencies). This assumption is true for a Fermi liquid because the imaginary part

---

[2]We omit spectral weights coming from physics or effects beyond the UV cutoff, such as those from interband transitions (or core electrons).

of the self-energy $\text{Im}\Sigma(\omega) \propto \omega^2 \to 0$ as $\omega \to 0$. However, this assumption does not hold for the power-law Green function in Eq. 3 because the Green function is not real when $\omega < \varepsilon_{\mathbf{p}}$.

A more general form [4,25] of Luttinger's theorem which does not require the Green function to be real (but is still based on a perturbative argument) is given by

$$n = \int \frac{d^d p}{(2\pi)^d} \frac{1}{\pi} \left( \phi_R(0) - \phi_R(-\infty) \right), \tag{5}$$

where $\phi_R(\omega)$ is the phase of the retarded Green function at frequency $\omega$. Since we are considering a system of spinless fermions, Eq. 5 does not have a factor of 2 in front, unlike Eq. 1. Notice that this equation reduces to Eq. 1 (without the spin degeneracy factor) when $\text{Im}\Sigma$ vanishes at $\omega = -\infty$ and $\omega = 0$. For the power-law Green function, we interpret $\omega = -\infty$ as the negative UV cutoff energy $-E$. Then, the phase of the retarded Green function at $\omega = -E$ is

$$\phi_R(-E) = \begin{cases} -\alpha\pi & \text{if } 0 < \alpha < 1, \\ -\alpha\pi + \pi & \text{if } 1 < \alpha < 2, \end{cases}$$

and the phase at $\omega = 0$ is

$$\phi_R(0) = \begin{cases} -\alpha\pi(1 - \theta(-\varepsilon_{\mathbf{p}})) & \text{if } 0 < \alpha < 1, \\ -\alpha\pi(1 - \theta(-\varepsilon_{\mathbf{p}})) + \pi & \text{if } 1 < \alpha < 2. \end{cases}$$

For both $0 < \alpha < 1$ and $1 < \alpha < 2$, one has $\phi_R(0) - \phi_R(-E) = \alpha\pi\theta(-\varepsilon_{\mathbf{p}})$. Consequently, Luttinger's theorem from Eq. 5 claims that the density

$$n = \alpha \int \frac{d^d p}{(2\pi)^d} \theta(-\varepsilon_{\mathbf{p}}). \tag{6}$$

This result is similar to that of Luttinger's for a Fermi liquid, $n = \int \frac{d^d p}{(2\pi)^d} \theta(-\tilde{\varepsilon}_{\mathbf{p}})$, with $\tilde{\varepsilon}_{\mathbf{p}}$ being the renormalized dispersion. The main difference is the prefactor $\alpha$ which comes from the fact that the Green function is complex.

## 2.2 Spectral function

To check the validity of Luttinger's theorem, one needs to know the density of the system. We begin by computing the spectral function which is equal to the discontinuity of the Green function across the branch cut,

$$
\begin{aligned}
A(\mathbf{p}, \omega) &= -\frac{1}{\pi} \text{Im} G(\mathbf{p}, \omega + i\eta) \\
&= -\frac{N}{2\pi i} \left[ \frac{1}{(\omega + i\eta - \varepsilon_{\mathbf{p}})^\alpha} - \frac{1}{(\omega - i\eta - \varepsilon_{\mathbf{p}})^\alpha} \right] \\
&= -\frac{N}{2\pi i} \frac{\theta(\varepsilon_{\mathbf{p}} - \delta - \omega)}{|\varepsilon_{\mathbf{p}} - \omega|^\alpha} \left( \frac{1}{e^{i\pi\alpha}} - \frac{1}{e^{-i\pi\alpha}} \right) \\
&= \frac{N \sin \pi\alpha}{\pi} \frac{\theta(\varepsilon_{\mathbf{p}} - \delta - \omega)}{|\varepsilon_{\mathbf{p}} - \omega|^\alpha}.
\end{aligned}
\tag{7}
$$

The normalization factor $N$ can be obtained from the spectral sum rule (Eq. 4). Substituting Eq. 7 into Eq. 4 and then solving for $N$, one finds that

$$
\begin{aligned}
N &= \frac{(1-\alpha)\pi}{\sin \pi\alpha} \frac{1}{(E + \varepsilon_{\mathbf{p}})^{1-\alpha} - \delta^{1-\alpha}} \\
&= \frac{(1-\alpha)\pi}{\sin \pi\alpha} \frac{1}{E^{1-\alpha} - \delta^{1-\alpha}}.
\end{aligned}
\tag{8}
$$

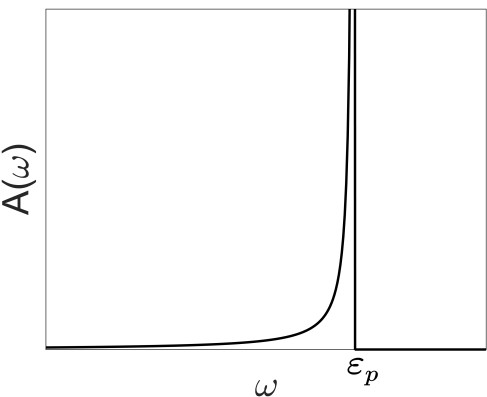

Figure 1: A plot of the spectral function $A(\omega)$ of fermions with power-law Green functions. The anomalous exponent $\alpha = 1.2$. Other values of $\alpha$ in the range $0 < \alpha < 2$ have the same qualitative behavior for $A(\omega)$.

Here, we have used the assumption $E \gg |\varepsilon_{\mathbf{p}}|$. This assumption is important for keeping $N$ independent of $\varepsilon_{\mathbf{p}}$. Note that the last line of Eq. 7 is positive even when $1 < \alpha < 2$, because $N$ is negative for such an $\alpha$. Explicitly, the final expression for $A(\mathbf{p}, \omega)$ is given by

$$A(\mathbf{p}, \omega) = \frac{|1 - \alpha|}{|E^{1-\alpha} - \delta^{1-\alpha}|} \frac{\theta(\varepsilon_{\mathbf{p}} - \delta - \omega)}{|\varepsilon_{\mathbf{p}} - \omega|^\alpha}. \tag{9}$$

Fig. 1 shows a plot of the spectral function for $\alpha = 1.2$. For a given momentum $\mathbf{p}$, there are excitations at all energies $\omega < \varepsilon_{\mathbf{p}}$. This behavior stems from our choice of the branch cut which lies along the negative real axis.

## 2.3 Occupation number

The occupation number in terms of $A(\mathbf{p}, \omega)$ is given by

$$n(\mathbf{p}) = \int d\omega \, n_f(\omega) A(\mathbf{p}, \omega), \tag{10}$$

where $n_f(\omega) \equiv \frac{1}{e^{\beta\omega} + 1}$ is the Fermi-Dirac distribution. The density of the system can then be calculated by integrating $n(\mathbf{p})$ over all momenta $\mathbf{p}$,

$$n = \int \frac{d^d p}{(2\pi)^d} n(\mathbf{p}). \tag{11}$$

At $T = 0$, the Fermi-Dirac distribution becomes a step function, $n_f(\omega) = \theta(-\omega)$. By inserting $1 = \theta(\varepsilon_{\mathbf{p}} - \delta) + \theta(-\varepsilon_{\mathbf{p}} + \delta)$ into the integrand of Eq. 10 and then integrating over $\omega$, we obtain

$$
\begin{aligned}
n(\mathbf{p}) &= \int_{-E}^{E} d\omega \, \theta(-\omega)[\theta(\varepsilon_{\mathbf{p}} - \delta) + \theta(-\varepsilon_{\mathbf{p}} + \delta)] A(\mathbf{p}, \omega) \\
&= \frac{\sin \pi\alpha}{\pi(1-\alpha)} N \theta(\varepsilon_{\mathbf{p}} - \delta)[(E + \varepsilon_{\mathbf{p}})^{1-\alpha} - \varepsilon_{\mathbf{p}}^{1-\alpha}] \\
&\quad + \frac{\sin \pi\alpha}{\pi(1-\alpha)} N \theta(-\varepsilon_{\mathbf{p}} + \delta)[(E + \varepsilon_{\mathbf{p}})^{1-\alpha} - \delta^{1-\alpha}].
\end{aligned}
$$

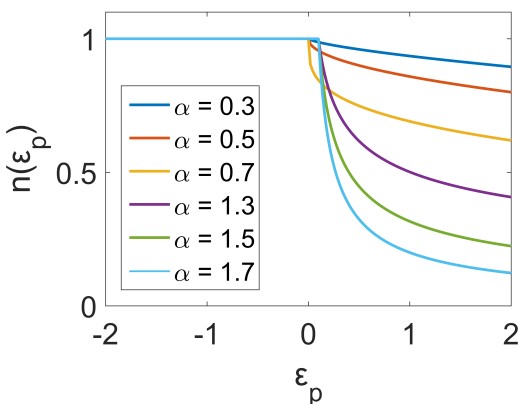

Figure 2: A plot of the occupation number $n(\mathbf{p})$ of fermions with power-law Green functions. The parameters used here are $E = 50$, and $\delta = 0.1$.

Finally, substituting $N$ from Eq. 8 into this equation and taking the limit $E \gg |\varepsilon_{\mathbf{p}}|$, one has

$$n(\mathbf{p}) = \theta(-\varepsilon_{\mathbf{p}} + \delta) + \theta(\varepsilon_{\mathbf{p}} - \delta)\frac{E^{1-\alpha} - \varepsilon_{\mathbf{p}}^{1-\alpha}}{E^{1-\alpha} - \delta^{1-\alpha}}. \tag{12}$$

For $0 < \alpha < 1$, setting $\delta = 0$, one obtains

$$n(\mathbf{p}) = \theta(-\varepsilon_{\mathbf{p}}) + \theta(\varepsilon_{\mathbf{p}})\left[1 - \left(\frac{\varepsilon_{\mathbf{p}}}{E}\right)^{1-\alpha}\right], \tag{13}$$

while for $1 < \alpha < 2$, taking the limits $E \gg \delta$ and $E \gg |\varepsilon_{\mathbf{p}}|$ gives

$$n(\mathbf{p}) = \theta(-\varepsilon_{\mathbf{p}} + \delta) + \theta(\varepsilon_{\mathbf{p}} - \delta)\left(\frac{\delta}{\varepsilon_{\mathbf{p}}}\right)^{\alpha-1}. \tag{14}$$

Fig. 2 shows a plot of the occupation number $n(\mathbf{p})$ for various values of $\alpha$. Since the occupation number is one for $\varepsilon_{\mathbf{p}} < 0$ and nonzero for $\varepsilon_{\mathbf{p}} > 0$, particle-hole symmetry is broken. This arises because the spectral function is nonzero only for energies below $\varepsilon_{\mathbf{p}}$.

### 2.4 Modified Luttinger count

In the case $0 < \alpha < 1$, using Eqs. 11 and 13, we find that the density is

$$n = \int \frac{d^d p}{(2\pi)^d}\theta(-\varepsilon_{\mathbf{p}}) + \int \frac{d^d p}{(2\pi)^d}\theta(\varepsilon_{\mathbf{p}})\left[1 - \left(\frac{\varepsilon_{\mathbf{p}}}{E}\right)^{1-\alpha}\right]. \tag{15}$$

Comparing this result to what is claimed by Luttinger's theorem in Eq. 6, one finds that the density obtained here is always greater than $\alpha \int \frac{d^d p}{(2\pi)^d}\theta(-\varepsilon_{\mathbf{p}})$. Consequently, Luttinger's theorem never holds for fermions with the power-law Green function when $0 < \alpha < 1$.

When $1 < \alpha < 2$, using Eqs. 11 and 14 gives the density

$$n = \int \frac{d^d p}{(2\pi)^d}\theta(-\varepsilon_{\mathbf{p}} + \delta) + \int \frac{d^d p}{(2\pi)^d}\theta(\varepsilon_{\mathbf{p}} - \delta)\left(\frac{\delta}{\varepsilon_{\mathbf{p}}}\right)^{\alpha-1}, \tag{16}$$

which in general differs from $\alpha \int \frac{d^d p}{(2\pi)^d}\theta(-\varepsilon_{\mathbf{p}})$. While Luttinger's theorem does not hold in general, we can still get Eq. 6 and Eq. 16 to agree by fine-tuning the energy function $\varepsilon_{\mathbf{p}}$,

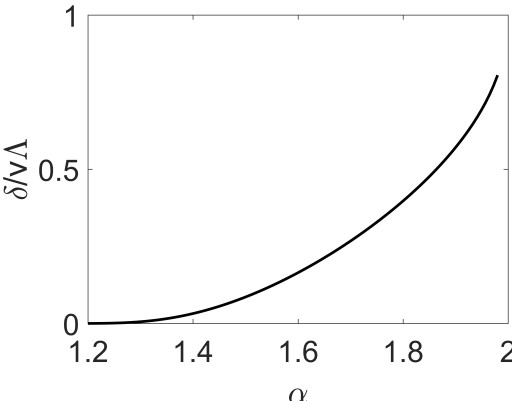

Figure 3: A plot of $\delta/v\Lambda$ vs. $\alpha$ obtained by solving Eq. 17 in the case $\delta < v\Lambda$. This shows the combination of parameters needed for Luttinger's theorem to be valid.

the exponent $\alpha$, and the cutoff $\delta$. For example, consider the case of a linear energy spectrum $\varepsilon_p = vp$ in one dimension, where the constant $v$ has units of velocity, and the momentum $p$ is chosen to be in the range $-\Lambda < p < \Lambda$. By equating Eq. 6 and Eq. 16, one can show that Luttinger's theorem holds when

$$(\alpha - 1)\Lambda = \frac{\delta}{v} + \frac{1}{2-\alpha}\left[\left(\frac{\delta}{v}\right)^{\alpha-1}\Lambda^{2-\alpha} - \frac{\delta}{v}\right]. \tag{17}$$

Numerically solving for the dimensionless ratio $\delta/v\Lambda$ as a function of $\alpha$ produces the result displayed in Fig. 3. When solving this equation, we require $\delta < v\Lambda$ to reflect the fact that $\delta$ is an infrared cutoff and thus must be smaller than other energy scales. For a given $\alpha$, the ratio $\delta/v\Lambda$ is fixed for Luttinger's theorem to be valid.

Although the above calculations are based on the spectral function having a sharp high-energy cutoff, our results regarding the validity of Luttinger's theorem remain unchanged even if we use a more general form of the cutoff (see Appendix A).

It is instructive to consider an alternate form of Green function in which the self-energy has a power-law form, $G(\mathbf{p}, \omega) \propto \frac{1}{\omega - \tilde{\varepsilon}_{\mathbf{p}} - \Sigma_{\text{PL}}(\mathbf{p}, \omega)}$ with $\Sigma_{\text{PL}}(\mathbf{p}, \omega) \propto \lambda \frac{(\omega - \varepsilon_{\mathbf{p}})^{\alpha}}{E^{\alpha-1}}$, where the dimensionless parameter $\lambda$ determines the strength of correlation of the self-energy. The advantage of this Green function over the power-law Green function (Eq. 3) is that it follows the canonical form of the Green function for an interacting system, i.e., $G \sim \frac{1}{\omega - \varepsilon_{\mathbf{p}} - \Sigma}$, and thus it is more physical than the power-law Green function. Nonetheless, this Green function reduces to the power-law Green function in the limit $\lambda \to \infty$. In Appendix B, we investigate the validity of Luttinger's theorem for this alternative Green function. We numerically compare both sides of Eq. 5. We find that, in general, Luttinger's theorem is not valid.

## 3 Luttinger's theorem for the spinless Luttinger liquid

Luttinger liquids are another fermionic system with a scale-invariant Green function. However, unlike the result we obtained above, Luttinger's theorem has been shown to be satisfied in Luttinger liquids [1,2]. To understand this discrepancy, we analytically verify Luttinger's theorem for a simple version of a Luttinger liquid, i.e., the spinless Luttinger model from Ref. [23]. The Hamiltonian of this model is given by

$$H = H_0 + H_I. \tag{18}$$

The non-interacting part of the Hamiltonian, $H_0$, is

$$H_0 = \sum_{\alpha=\pm} \sum_{|p-\alpha p_f|<\Lambda} v_f(\alpha p - p_f)c_{\alpha,p}^\dagger c_{\alpha,p}, \tag{19}$$

where $\alpha = +$ denotes the right-moving fermions (right movers) and $\alpha = -$ denotes the left-moving fermions (left movers). The operators $c_{\alpha,p}^\dagger$ and $c_{\alpha,p}$ are the fermion creation and annihilation operators in momentum space, respectively. (In real space, we denote the fermions by $\psi_\alpha(x)$ and $\psi_\alpha^\dagger(x)$.) Also, $v_f$ and $p_f$ denote the Fermi velocity and Fermi momentum of the non-interacting system, respectively. The momentum cutoff $\Lambda$ is chosen such that, in the momentum range $-\Lambda < p - \alpha p_f < \Lambda$, the non-interacting dispersion is linear. The fermion-fermion interaction, $H_I$, is given by[3]

$$H_I = \int dx \int dx' \frac{1}{2} V(x-x')$$
$$\times \left[ \rho_+(x)\rho_+(x') + \rho_-(x)\rho_-(x') + \rho_+(x)\rho_-(x') + \rho_-(x)\rho_+(x') \right], \tag{20}$$

where $\rho_\alpha(x) \equiv \psi_\alpha^\dagger(x)\psi_\alpha(x)$ is the density of fermions in branch $\alpha$ at point $x$. The first two terms are the interactions between fermions from the same branch. They are known as the $g_4$ process [27]. The last two terms represent the inter-branch interactions or the $g_2$ process [27]. For a system of spin-1/2 fermions, there is also an interaction between two branches with their spins exchanged or the $g_1$ process [27]. In the spinless system, $g_1$ is the same as $g_2$. In general, $g_2$ and $g_4$ can have different interaction strengths, but the form of $H_I$ we consider in Eq. 20 has $g_4 = g_2 = V$.

In this section, we investigate Luttinger's theorem for the right-moving branch with $\alpha = +$. The conclusion we have for the right-movers should also be applicable to the left-movers. As in the case of the power-law Green function, we calculate the density of fermions and compare it with Luttinger's theorem. The starting point is the spectral function of this model [23],

$$A_+(p,\omega) = \frac{1}{\gamma\Gamma^2(\gamma)} \left( \frac{r}{2\tilde{v}_f} \right)^{2\gamma} \left[ \theta(\omega - \tilde{v}_f|p|)(\omega + \tilde{v}_f p)^\gamma(\omega - \tilde{v}_f p)^{\gamma-1} e^{-\frac{\omega r}{\tilde{v}_f}} \right.$$
$$\times \Phi\left( 1, 1+\gamma, \frac{r}{2\tilde{v}_f}(\omega + \tilde{v}_f p) \right)$$
$$+ \theta(-\omega - \tilde{v}_f|p|)(-\omega - \tilde{v}_f p)^\gamma(-\omega + \tilde{v}_f p)^{\gamma-1} e^{\frac{\omega r}{\tilde{v}_f}}$$
$$\left. \times \Phi\left( 1, 1+\gamma, \frac{r}{2\tilde{v}_f}(-\omega - \tilde{v}_f p) \right) \right], \tag{21}$$

where $\Gamma(x)$ is the gamma function, $\Phi(a,b,x)$ denotes the confluent hypergeometric function[4], $\tilde{v}_f \equiv v_f \left( 1 + \frac{V(q=0)}{\pi v_f} \right)^{1/2}$ is the renormalized velocity, $r$ is the interaction range, and $\gamma$ determines the interaction strength. The precise definitions of $r$ and $\gamma$ are given in Ref. [23]. Here, the momentum $p$ is measured with respect to the Fermi point, and thus the total momentum is $p + p_f$. We note that $\Phi(a,b,0) = 1$. As a result, in the short interaction range limit, $r \to 0$, the spectral function has a scaling form. However, at large $\omega$, $A_+(p,\omega) \sim \omega^{2\gamma-1}$ which violates the sum rule for $\gamma > 0$. To avoid this problem, we keep $r$ finite as a regulator throughout the calculation. The plot of $A_+(p,\omega)$ from Eq. 21 is displayed in Fig. 4.

---

[3]One can show that this is the same interaction as Ref. [23] by transforming to a bosonic basis [23, 26, 27], $b_p^\dagger = \left( \frac{2\pi}{L|p|} \right)^{\frac{1}{2}} \sum_{\alpha=\pm} \theta(\alpha p)\rho_\alpha(-p)$ and $b_p = \left( \frac{2\pi}{L|p|} \right)^{\frac{1}{2}} \sum_{\alpha=\pm} \theta(\alpha p)\rho_\alpha(p)$.

[4]Other notations [28] of the confluent hypergeometric function are $M(a,b,x)$ and $_1F_1(a,b,x)$.

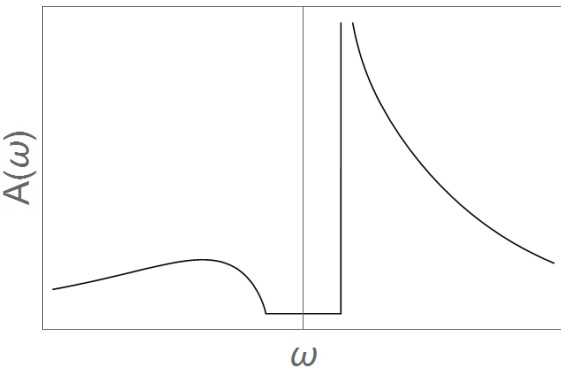

Figure 4: The plot of the spectral function $A_+(\omega)$. The parameters used to generate the plot are $p = 3$, $r = 0.2$, $\tilde{v}_f = 1$, and $\gamma = 0.8$.

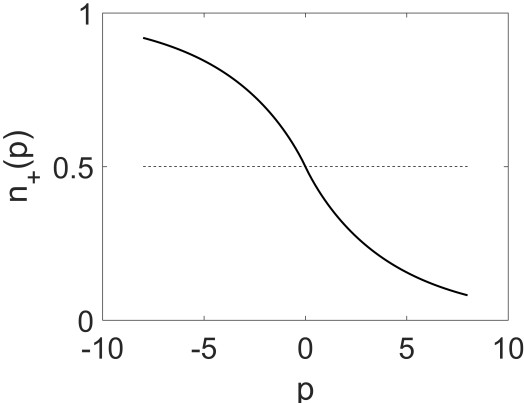

Figure 5: The plot of $n_+$ vs $p$. The parameters used to generate the plot are $r = 0.2$, $\tilde{v}_f = 1$, and $\gamma = 0.8$.

One can calculate the occupation number of the right movers at $T = 0$ as

$$n_+(p) = \int_{-\infty}^{\infty} d\omega\, n_F(\omega) A_+(p, \omega), \tag{22}$$

where $n_F(\omega) = \theta(-\omega)$ is the Fermi-Dirac distribution at $T = 0$. The plot of $n_+(p)$ is shown in Fig. 5. The important feature of $n_+(p)$ is that it is an odd function with respect to $n_+ = 1/2$ (see Appendix D). This is a signature that the system has particle-hole symmetry. Based on this property, the density of the right-movers at $T = 0$ can be computed as

$$n_+ = \int_{-\Lambda}^{\Lambda} \frac{dp}{2\pi} n_+(p) = \frac{\Lambda}{2\pi}. \tag{23}$$

From the spectral function, one can obtain the real and imaginary parts of the retarded Green function $G_+^R$ (see Appendix E). We find that $G_+^R$ is real at $\omega = 0$ and $\omega = -\infty$. Furthermore, at $\omega = 0$, $G_+^R$ becomes positive when $p < 0$. This means that Luttinger's theorem for the spinless Luttinger model has the standard form of Eq. 1 (without the spin degeneracy factor). It can also be written as

$$n_+ = \int \frac{dp}{2\pi} \theta(-p), \tag{24}$$

which counts only states below $p_f$. To be consistent with the density calculation, the range of $p$ in the momentum integral is $-\Lambda < p < \Lambda$. The integral can then be evaluated as

$$\int_{-\Lambda}^{\Lambda} \frac{dk}{2\pi} \theta(-p) = \frac{\Lambda}{2\pi}. \tag{25}$$

The agreement between Eq. 23 and Eq. 25 means that Luttinger's theorem holds for the right-moving branch of the spinless Luttinger liquid.

Two properties of this model are important for the Luttinger sum rule to be valid. First, the Luttinger sum rule of this model can be simplified to the traditional form. This result stems from the fact that $G_+^R(p, \omega)$ is real at frequencies $\omega = 0$ and $\omega = -\infty$ and $G_+^R(p, \omega)$ changes sign at the momentum $p_f$. The second property is particle-hole symmetry. This leads to the result that the fermion density equals the number of states below $p_f$. Combining these two properties, it is obvious that Luttinger's theorem holds in the spinless Luttinger model. From the discussion in Section 2.1 and Fig. 2, it is clear that fermions with power-law Green functions do not satisfy either of these properties.

# 4 Discussion and Conclusion

The key result of this paper is that, in general, Luttinger's theorem is not valid for fermions with power-law Green functions. However, one cannot conclude whether Luttinger's theorem holds for a fermionic system based solely on the fact that its Green function satisfies a scaling form (Eq. 2). Further constraints are required. A Luttinger liquid is one example in which the Green function is scale invariant but Luttinger's theorem is satisfied.

The two properties we mentioned at the end of previous section, i.e., the vanishing of $\text{Im}\,G(\omega)$ at $\omega = 0, -\infty$ and particle-hole symmetry, are also necessary for Luttinger's theorem to be valid in the Hubbard model [8] and the $SU(N)$ Hubbard model [11]. This indicates that these properties are important for the validity of Luttinger's theorem in a fermionic system. One needs to keep in mind that there exist special cases in which neither property is present, but Luttinger's theorem is still valid. We find one such case in this work: a system of fermions with the power-law Green function and the exponent $\alpha$ in the range $1 < \alpha < 2$. A simpler example is a system of noninteracting fermions away from half-filling. In this case, Luttinger's theorem is valid but the system is clearly not particle-hole symmetric. Hence, we conjecture that particle-hole symmetry and $\text{Im}\,G(\omega = 0, -\infty) = 0$ are sufficient but not necessary conditions for the validity of Luttinger's theorem. A rigorous proof is necessary to establish that these properties are the general criteria for deciding which system respects Luttinger's theorem.

### Acknowledgments

We thank NSF DMR-1461952 for partial funding of this project. KL is supported by the Department of Physics at the University of Illinois and a scholarship from the Ministry of Science and Technology, Royal Thai Government. ZL is supported by the Department of Physics at the University of Illinois and a scholarship from the Agency of Science, Technology and Research, Singapore.

# A Modified Luttinger count with generalized cutoff

Let us consider the spectral function of the form

$$A(p,\omega) = \begin{cases} N\frac{\sin \pi \alpha}{\pi}\frac{\theta(\varepsilon_p - \delta - \omega)}{|\varepsilon_p - \omega|^\alpha}, & \omega > -E, \\ N\frac{\sin \pi \alpha}{\pi}\frac{f(\omega)}{E^\alpha}, & \omega < -E, \end{cases}$$

where $f(\omega)$ is a dimensionless cutoff function. There are two restrictions that one needs to put on $f(\omega)$. First, $f(\omega)$ must fall off faster than $\omega^{-1}$ as $\omega \to \pm\infty$ in order for the integral $\int A(\omega)d\omega$ to converge. Second, the integral $\int_{-\infty}^{-E}\frac{f(\omega)}{E}d\omega \ll 1$. With this requirement, the spectral weight from the cutoff function is much less than the total spectral weight, i.e. $\int_{-\infty}^{E} N\frac{\sin \pi \alpha}{\pi}\frac{f(\omega)}{E^\alpha}d\omega \ll \int_{-\infty}^{\infty} A(\omega)d\omega = 1$. Here, we explicitly exclude the physics or effects from energies beyond $\pm E$, for example, interband transitions (from core electrons).

Using Eq. 4, one finds that the normalization factor in the limit $E \gg |\varepsilon_{\mathbf{p}}|$ is given by

$$N = \frac{(1-\alpha)\pi}{\sin \pi \alpha}\frac{1}{E^{1-\alpha} - \delta^{1-\alpha} + cE^{1-\alpha}}, \tag{26}$$

where $c \equiv \int_{-\infty}^{-E}\frac{f(\omega)}{E}d\omega$ is a small parameter. Following the same procedure as in Section 2.3, one finds the occupation number is

$$n(\mathbf{p}) = \theta(-\varepsilon_{\mathbf{p}} + \delta) + \theta(\varepsilon_{\mathbf{p}} - \delta)\frac{E^{1-\alpha} - \varepsilon_{\mathbf{p}}^{1-\alpha} + (1-\alpha)E^{1-\alpha}c}{E^{1-\alpha} - \delta^{1-\alpha} + (1-\alpha)E^{1-\alpha}c}. \tag{27}$$

For $0 < \alpha < 1$, setting $\delta = 0$, one obtains

$$
\begin{aligned}
n(\mathbf{p}) &= \theta(-\varepsilon_{\mathbf{p}}) + \theta(\varepsilon_{\mathbf{p}})\left[1 - \frac{1}{1+(1-\alpha)c}\left(\frac{\varepsilon_{\mathbf{p}}}{E}\right)^{1-\alpha}\right] \\
&\approx \theta(-\varepsilon_{\mathbf{p}}) + \theta(\varepsilon_{\mathbf{p}})\left[1 - \left(\frac{\varepsilon_{\mathbf{p}}}{E}\right)^{1-\alpha} + (1-\alpha)\left(\frac{\varepsilon_{\mathbf{p}}}{E}\right)^{1-\alpha}c\right],
\end{aligned}
\tag{28}
$$

while for $1 < \alpha < 2$, taking the limits $E \gg \delta$ and $E \gg \varepsilon_p$ gives

$$
\begin{aligned}
n(\mathbf{p}) &= \theta(-\varepsilon_{\mathbf{p}} + \delta) + \theta(\varepsilon_{\mathbf{p}} - \delta)\frac{\left(\frac{\delta}{\varepsilon_{\mathbf{p}}}\right)^{\alpha-1} - (1-\alpha)\left(\frac{\delta}{E}\right)^{\alpha-1}c}{1 - (1-\alpha)\left(\frac{\delta}{E}\right)^{\alpha-1}c} \\
&\approx \theta(-\varepsilon_{\mathbf{p}} + \delta) + \theta(\varepsilon_{\mathbf{p}} - \delta)\left[\left(\frac{\delta}{\varepsilon_{\mathbf{p}}}\right)^{\alpha-1} - \left(1 - \left(\frac{\delta}{\varepsilon_{\mathbf{p}}}\right)^{\alpha-1}\right)(1-\alpha)\left(\frac{\delta}{E}\right)^{\alpha-1}c\right]. 
\end{aligned}
\tag{29}
$$

For $0 < \alpha < 1$, the density is then given by

$$n = \int \frac{d^d p}{(2\pi)^d}\theta(-\varepsilon_{\mathbf{p}}) + \int \frac{d^d p}{(2\pi)^d}\theta(\varepsilon_{\mathbf{p}})\left[1 - \left(\frac{\varepsilon_{\mathbf{p}}}{E}\right)^{1-\alpha} + (1-\alpha)\left(\frac{\varepsilon_{\mathbf{p}}}{E}\right)^{1-\alpha}c\right], \tag{30}$$

and, for $1 < \alpha < 2$, the density is

$$
\begin{aligned}
n &= \int \frac{d^d p}{(2\pi)^d}\theta(-\varepsilon_{\mathbf{p}} + \delta) \\
&\quad + \int \frac{d^d p}{(2\pi)^d}\theta(\varepsilon_{\mathbf{p}} - \delta)\left[\left(\frac{\delta}{\varepsilon_{\mathbf{p}}}\right)^{\alpha-1} - \left(1 - \left(\frac{\delta}{\varepsilon_{\mathbf{p}}}\right)^{\alpha-1}\right)(1-\alpha)\left(\frac{\delta}{E}\right)^{\alpha-1}c\right]. 
\end{aligned}
\tag{31}
$$

For the general high-energy cutoff, the claim of Luttinger's theorem is modified from Eq. 6. Since the phase of the retarded Green function at infinity is bounded as $-\pi < \phi_R(-\infty) \leq \pi$, Luttinger's theorem claims that the particle density for $0 < \alpha < 1$ is bounded above:

$$n \quad < \quad (1-\alpha) \int \frac{d^d p}{(2\pi)^d} \theta\left(\varepsilon_{\mathbf{p}}\right) + \int \frac{d^d p}{(2\pi)^d} \theta\left(-\varepsilon_{\mathbf{p}}\right).$$

Since the coefficient in front of the $\theta\left(\varepsilon_{\mathbf{p}}\right)$ integral is less than one, Luttinger's theorem undercounts the particle density for $\varepsilon_{\mathbf{p}}$ just above the Fermi level, or more precisely when $\left(\frac{\varepsilon_{\mathbf{p}}}{E}\right)^{1-\alpha} < \alpha$. Similarly for $1 < \alpha < 2$, the particle density according to Luttinger's theorem is bounded as

$$n \quad < \quad (2-\alpha) \int \frac{d^d p}{(2\pi)^d} \theta\left(\varepsilon_{\mathbf{p}}\right) + 2 \int \frac{d^d p}{(2\pi)^d} \theta\left(-\varepsilon_{\mathbf{p}}\right).$$

Since the coefficient $2-\alpha < 1$, if the energy spectrum is such that $\varepsilon_{\mathbf{p}} > \delta$ and $\left(\frac{\delta}{\varepsilon_{\mathbf{p}}}\right)^{\alpha-1} > 2-\alpha$, then Luttinger's theorem does not hold. Therefore, we reach the same conclusion as the sharp cutoff case ($f(\omega) = 0$). Luttinger's theorem is not valid in general; only for some specific values of parameters can Luttinger's theorem hold.

# B   Luttinger's theorem for Fermions with power-law self-energy

In this Appendix, we examine the validity of Luttinger's theorem for fermions with Green function of the form,

$$G(\mathbf{p}, \omega) = \frac{N}{\omega - \tilde{\varepsilon}_{\mathbf{p}} - \Sigma_{\text{PL}}(\mathbf{p}, \omega)}, \tag{32}$$

where the self-energy is given by

$$\Sigma_{\text{PL}}(\mathbf{p}, \omega) = -\text{sgn}(1-\alpha)\lambda \frac{(\omega - \varepsilon_{\mathbf{p}})^\alpha}{E^{\alpha-1}}. \tag{33}$$

Here, $N$ is the normalization factor, $E$ is the cutoff energy discussed in Section 2, $\lambda > 0$ is a dimensionless coefficient which determines the correlation strength of this self-energy, and the sign function, $\text{sgn}(1-\alpha)$, in front keeps the imaginary part of the self-energy to be negative for $\alpha$ in the range $0 < \alpha < 2$. The branch cut from the term $(\omega - \varepsilon_{\mathbf{p}})^\alpha$ is chosen to lie along the negative real axis and the phase is defined to be in the range $-\pi < \phi \leq \pi$. In the limit $\lambda \to \infty$, this Green function reduces to the power-law Green function (Eq. 3) we investigate in the main text. For simplicity of the calculation we set $\tilde{\varepsilon}_{\mathbf{p}} = \varepsilon_{\mathbf{p}}$. As in Section 2, we assume the cutoff energy, $E$, to be much larger than the energy function $\varepsilon_{\mathbf{p}}$. This assumption allows the normalization factor, $N$, to be momentum independent. In the calculation below, we only consider the case of $0 < \alpha < 1$. For the case $1 < \alpha < 2$, one needs to include a cutoff at low energy, $\delta$, in $\Sigma_{PL}$ to regulate the infrared divergence.

To verify Luttinger's theorem, one needs to compare both sides of Eq. 5,

$$n = \int \frac{d^d p}{(2\pi)^d} \frac{1}{\pi} \left(\phi_R(0) - \phi_R(-\infty)\right). \tag{34}$$

We start by discussing how one can calculate the density of fermions, $n$, from this Green

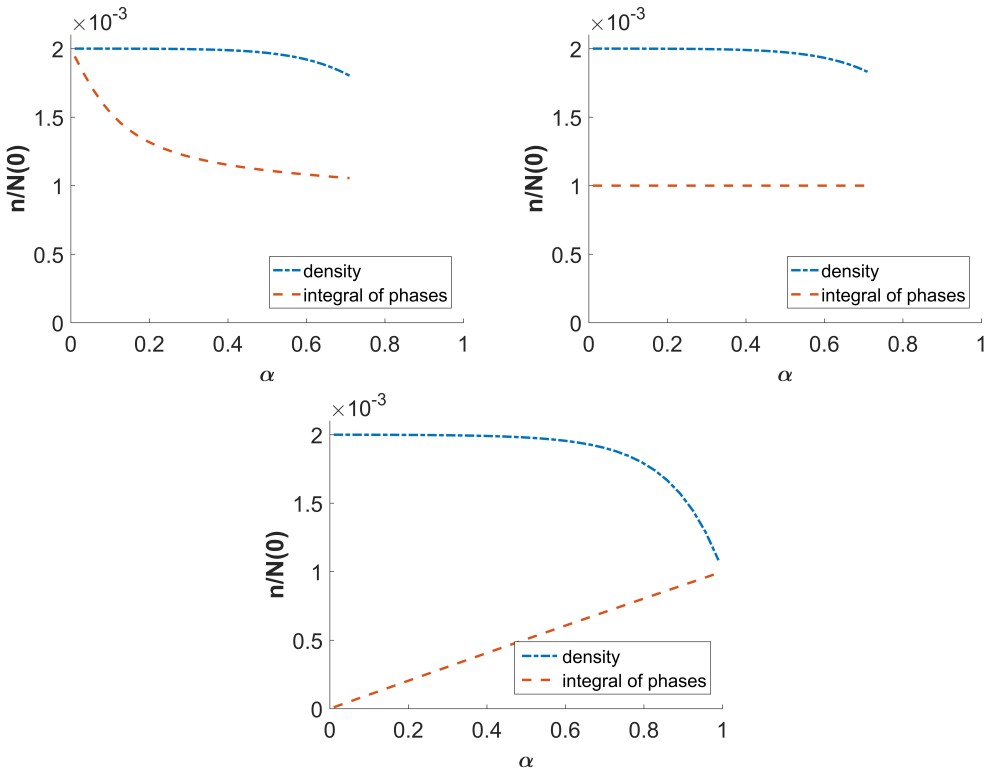

Figure 6: Comparison plots of the fermion density and the integral of the phases for $0 < \alpha < 1$ with (a) $\lambda = 0.7$, (b) $\lambda = 1$, and (c) $\lambda = 100$. Other parameters used in generating these plots are $E = 1$ and $W = 0.001E$. With these parameters and $\alpha$ in the range shown in the plots, Eq. 37 is satisfied. $n$ in $n/N(0)$ labeled on the vertical axis is a nominal symbol for the fermion density or the integral of the phases.

function. The spectral function of this Green function is given by

$$A(\mathbf{p}, \omega) = -\frac{1}{\pi} \text{Im} G(\mathbf{p}, \omega)$$

$$= \frac{N \text{sgn}(1-\alpha) \frac{\lambda|\omega - \varepsilon_{\mathbf{p}}|^{\alpha}}{\pi E^{\alpha-1}} \sin\left(\theta(-\omega + \varepsilon_{\mathbf{p}})\pi\alpha\right)}{\left(\omega - \varepsilon_{\mathbf{p}} + \text{sgn}(1-\alpha) \frac{\lambda|\omega - \varepsilon_{\mathbf{p}}|^{\alpha}}{E^{\alpha-1}} \cos\left(\theta(-\omega + \varepsilon_{\mathbf{p}})\pi\alpha\right)\right)^2 + \left(\frac{\lambda|\omega - \varepsilon_{\mathbf{p}}|^{\alpha}}{E^{\alpha-1}} \sin\left(\theta(-\omega + \varepsilon_{\mathbf{p}})\pi\alpha\right)\right)^2}.$$
(35)

One can normalize the spectral function by using Eq. 4. We note that in the limit $E \gg |\varepsilon_{\mathbf{p}}|$, one can set $\varepsilon_{\mathbf{p}} = 0$ in the normalization factor. The occupation number can then be computed from Eq. 10. Finally, using Eq. 11, we obtain the density, $n$. In order to perform the integral in Eq. 11, we replace the momentum integral, $\int \frac{d^d p}{(2\pi)^d}$, with the energy integral, $N(0) \int_{-W}^{W} d\varepsilon$. Here, $W$ is a bandwidth of the dispersion $\varepsilon_{\mathbf{p}}$ (this means $W \sim |\varepsilon_{\mathbf{p}}| \ll E$) and $N(0)$ is the density of state which is assumed to be constant. The numerical results of the density are plotted for $0 < \alpha < 1$ in Fig. 6.

We now calculate the integral of the phases on the right hand side of Eq. 34. At $\omega = 0$,

we have

$$
\begin{aligned}
G(\mathbf{p}, 0) &= \frac{N}{-\varepsilon_\mathbf{p} + \lambda \frac{(-\varepsilon_\mathbf{p})^\alpha}{E^{\alpha-1}}} \\
&= \frac{N E^{\alpha-1}}{\lambda(-\varepsilon_\mathbf{p})^\alpha}.
\end{aligned} \tag{36}
$$

In the second line, we can drop the $-\varepsilon_\mathbf{p}$ term because the denominator is dominated by the self-energy term (we have here $0 < \alpha < 1$ and $E \gg |\varepsilon_\mathbf{p}|$). More specifically, one needs

$$
\lambda \left( \frac{E}{|\varepsilon_\mathbf{p}|} \right)^{1-\alpha} \gg 1. \tag{37}
$$

This means

$$
\phi_R(0) = -\alpha \pi (1 - \theta(-\varepsilon_\mathbf{p})). \tag{38}
$$

As in the main text, we interpret $-\infty$ in $\phi_R(-\infty)$ as the negative cutoff energy, $\omega = -E$. In the limit $E \gg |\varepsilon_\mathbf{p}|$,

$$
\begin{aligned}
G(\mathbf{p}, -E) &= \frac{1}{-E + \frac{\lambda}{E^{\alpha-1}}(-E)^\alpha} \\
&= \frac{1}{E\left((-1 + \lambda \cos \pi\alpha) + i\lambda \sin \pi\alpha\right)}.
\end{aligned} \tag{39}
$$

Hence,

$$
\phi_R(-E) = \begin{cases} -\arctan\left(\frac{\lambda \sin \pi\alpha}{-1 + \lambda \cos \pi\alpha}\right) & \text{if } -1 + \lambda \cos \pi\alpha > 0 \\ -\arctan\left(\frac{\lambda \sin \pi\alpha}{-1 + \lambda \cos \pi\alpha}\right) - \pi & \text{if } -1 + \lambda \cos \pi\alpha < 0. \end{cases}
$$

We note that in the limit $\lambda \to \infty$, $\phi_R(-E) \to -\pi\alpha$ as we expect from Section 2. The momentum integral on the right hand side of Eq. 34 can be converted to the energy integral in the same way as in the density calculation above. The numerical result of the integral of the phases for $0 < \alpha < 1$ is displayed in Fig. 6.

In Fig. 6, we plot the fermion density (left hand side of Eq. 34) and the integral of the phases (right hand side of Eq. 34) as a function of $\alpha$ for $0 < \alpha < 1$. The numerical integration of the phase terms is not stable for small $\lambda$ and $\alpha \gtrsim 0.7$. Hence, we only display the plots in Figs. 6(a) and 6(b) with $\alpha$ in the range $0 < \alpha < 0.7$. One can see that, in general, Luttinger's theorem does not hold for this system. Furthermore, when $\lambda \to \infty$ and $\alpha \to 1$, the Green function represents a normal fermion and we recover Luttinger's theorem as shown in Fig. 6(c).

## C  Spectral sum rule of $A_+(p, \omega)$

In this Appendix, we verify that the spectral function $A_+(p, \omega)$ satisfies the spectral sum rule (Eq. 4). This will lead to an equation useful in the density calculation in Appendix D. We note that the confluent hypergeometric function $\Phi(1, 1 + \alpha, z)$ and the incomplete gamma function $\gamma(\alpha, z) \equiv \int\limits_0^z dt \, t^{\alpha-1} e^{-t}$ are related [28] through

$$
\Phi(1, 1 + \alpha, z) = \alpha e^z z^{-\alpha} \gamma(\alpha, z). \tag{40}
$$

Applying this relation to Eq. 21, one obtains the spectral sum

$$\int_{-\infty}^{\infty} A_+(p,\omega)d\omega = \frac{1}{\Gamma^2(\gamma)}\left(\frac{r}{2\tilde{v}_f}\right)^{\gamma}\int_{\tilde{v}_f|p|}^{\infty} d\omega\left[e^{-\frac{r}{2\tilde{v}_f}(\omega-p\tilde{v}_f)}(\omega-\tilde{v}_fp)^{\gamma-1}\gamma\left(\gamma,\frac{r}{2\tilde{v}_f}(\omega+p\tilde{v}_f)\right)\right.$$
$$\left. +e^{-\frac{r}{2\tilde{v}_f}(\omega+p\tilde{v}_f)}(\omega+\tilde{v}_fp)^{\gamma-1}\gamma\left(\gamma,\frac{r}{2\tilde{v}_f}(\omega-p\tilde{v}_f)\right)\right]. \quad (41)$$

Since the integral for the case $p > 0$ is the same as the integral for the case $p < 0$, we can set $p > 0$ without loss of generality. We perform a change of variables, $\omega' = \omega - p\tilde{v}_f$ on the first term and $\omega' = \omega + p\tilde{v}_f$ in the second term of the integrand. We then let $\omega' = \frac{2\tilde{v}_f}{r}x$. Substituting in the integral representation of $\gamma(\alpha,z)$, we have

$$\int_{-\infty}^{\infty} A_+(p,\omega)d\omega = \frac{1}{\Gamma^2(\gamma)}\left(\int_0^{\infty} dx\int_0^{x+pr} dt + \int_{pr}^{\infty} dx\int_0^{x-pr} dt\right)e^{-x}x^{\gamma-1}e^{-t}t^{\gamma-1}.$$

We can show that the spectral sum is 1 by splitting the limits as

$$\int_{-\infty}^{\infty} A_+(p,\omega)d\omega = \frac{2}{\Gamma^2(\gamma)}\int_0^{\infty} dx\int_0^x dt\,e^{-x}x^{\gamma-1}e^{-t}t^{\gamma-1}$$
$$+\frac{1}{\Gamma^2(\gamma)}\left(\int_0^{\infty} dx\int_x^{x+pr} dt + \int_{pr}^{\infty} dx\int_{x-pr}^x dt + \int_0^{pr} dx\int_0^x dt\right)e^{-x}x^{\gamma-1}e^{-t}t^{\gamma-1}. \quad (42)$$

The first term on the right hand side of Eq. 42 can be written in terms of $\gamma(\gamma,x)$ as $\frac{2}{\Gamma^2(\gamma)}\int_0^{\infty} dx\,e^{-x}x^{\gamma-1}\gamma(\gamma,x)$. Using the integral formula of the incomplete gamma function [28], $\int_0^{\infty} dx\,x^{a-1}e^{-sz}\gamma(b,x) = \frac{\Gamma(a+b)}{b(1+s)^{a+b}}F(1,a+b;1+b;\frac{1}{1+s})$ where Re $s > 0$ and Re$(a+b) > 0$, we find

$$\frac{2}{\Gamma^2(\gamma)}\int_0^{\infty} dx\,e^{-x}x^{\gamma-1}\gamma(\gamma,x) = \frac{2^{1-2\gamma}\Gamma(2\gamma)}{\gamma\Gamma^2(\gamma)}F(1,2\gamma;1+\gamma;\frac{1}{2}).$$

Here, $F(a,b;c;z)$ denotes the hypergeometric function. Applying the identities [28]

$$F(a,b;\frac{1}{2}(a+b)+\frac{1}{2},\frac{1}{2}) = \frac{\sqrt{\pi}\Gamma(\frac{1}{2}(a+b)+\frac{1}{2})}{\Gamma(\frac{1}{2}a+\frac{1}{2})\Gamma(\frac{1}{2}b+\frac{1}{2})} \quad (43)$$

and

$$\Gamma(2z) = \frac{1}{\sqrt{\pi}}2^{2z-1}\Gamma(z)\Gamma(z+\frac{1}{2}), \quad (44)$$

with $2z \neq 0,-1,-2,...$, one finds

$$\frac{2}{\Gamma^2(\gamma)}\int_0^{\infty} dx\int_0^x dt\,e^{-x}x^{\gamma-1}e^{-t}t^{\gamma-1} = 1.$$

We next show that the second term on the right hand side of Eq. 42 vanishes. Let us define a function $I(a)$ by

$$I(a) \equiv \frac{1}{\Gamma^2(\gamma)} \left( \int\limits_0^\infty dx \int\limits_x^{x+a} dt + \int\limits_a^\infty dx \int\limits_{x-a}^x dt + \int\limits_0^a dx \int\limits_0^x dt \right) e^{-x} x^{\gamma-1} e^{-t} t^{\gamma-1}. \quad (45)$$

The second term on the right hand side of Eq. 42 can be written as $I(pr)$. We note that $I(0) = 0$ and, by using the fundamental theorem of calculus, $I'(a) = 0$. This means $I(a) = 0$ for any $a$ and thus the second term on the right hand side of Eq. 42, $I(pr)$, equals zero. Consequently, the spectral sum $\int\limits_{-\infty}^\infty A_+(p_f + p, \omega) d\omega = 1$. Substituting Eq. 21 into the spectral sum rule (Eq. 4), we have

$$1 = \frac{1}{\gamma \Gamma^2(\gamma)} \left( \frac{r}{2\tilde{v}_f} \right)^{2\gamma} \int\limits_{\tilde{v}_f |p|}^\infty d\omega (\omega + \tilde{v}_f p)^\gamma (\omega - \tilde{v}_f p)^{\gamma-1} e^{-\frac{\omega r}{\tilde{v}_f}} \Phi(1, 1+\gamma, \frac{r}{2\tilde{v}_f} (\omega + \tilde{v}_f p))$$

$$+ \frac{1}{\gamma \Gamma^2(\gamma)} \left( \frac{r}{2\tilde{v}_f} \right)^{2\gamma} \int\limits_{\tilde{v}_f |p|}^\infty d\omega (\omega - \tilde{v}_f p)^\gamma (\omega + \tilde{v}_f p)^{\gamma-1} e^{\frac{\omega r}{\tilde{v}_f}} \Phi(1, 1+\gamma, \frac{r}{2\tilde{v}_f} (\omega - \tilde{v}_f p)). \quad (46)$$

This equation is important for the density calculation in Appendix D.

## D Occupation number and density of the spinless Luttinger liquid at $T = 0$

The occupation number of the right movers in a momentum state $p$ is given by

$$n_+(p) = \int\limits_{-\infty}^\infty d\omega n_F(\omega) A_+(p, \omega), \quad (47)$$

where $n_F(\omega)$ is the Fermi-Dirac distribution. Substituting Eq. 21 into Eq. 47 and taking the zero temperature limit (so $n_F(\omega) = \theta(-\omega)$), we have

$$n_+(p) = \frac{1}{\gamma \Gamma^2(\gamma)} \left( \frac{r}{2\tilde{v}_f} \right)^{2\gamma} \int\limits_{\tilde{v}_f |p|}^\infty d\omega (\omega - \tilde{v}_f p)^\gamma (\omega + \tilde{v}_f p)^{\gamma-1} e^{-\frac{\omega r}{\tilde{v}_f}} \Phi(1, 1+\gamma, \frac{r}{2\tilde{v}_f} (\omega - \tilde{v}_f p)). \quad (48)$$

Using Eq. 46, we can show that

$$n_+(-p) = 1 - n_+(p) \quad (49)$$

or

$$n+(-p) - \frac{1}{2} = -\left( n+(p) - \frac{1}{2} \right). \quad (50)$$

This means $n(p) - \frac{1}{2}$ is an odd function. Using Eq. 49, one can show that the density of the right mover is

$$n_+ = \int\limits_{-\Lambda}^\Lambda n_+(p) \frac{dp}{2\pi} = \frac{\Lambda}{2\pi}. \quad (51)$$

# E   Luttinger's theorem of the spinless Luttinger liquid

We determine the form of Luttinger's theorem for a spinless Luttinger liquid. From Eq. 5, we need to know the phases of the retarded Green function $\phi_R$ at $\omega = 0$ and $\omega = -\infty$. For fermions, in the limit $\omega \to -\infty$, the retarded Green function $G^R(\omega) \to \frac{1}{\omega}$. This means $\phi_R(-\infty) = -\pi$.

We next calculate $\phi_R(0)$. The imaginary part of the retarded Green function is related to the spectral function by $\mathrm{Im}G^R(\omega) = -\pi A(\omega)$. Substituting in $A_+(\omega)$ from Eq. 21, we find

$$
\mathrm{Im}G_+^R(\omega = 0) = -\frac{\pi}{\gamma\Gamma^2(\gamma)}\theta(-\tilde{v}_f|p|)\left(\frac{r}{2\tilde{v}_f}\right)^{2\gamma}
$$
$$
\times\left((\tilde{v}_f p)^\gamma(-\tilde{v}_f p)^{\gamma-1}\Phi(1, 1+\gamma, \frac{pr}{2}) + (-\tilde{v}_f p)^\gamma(\tilde{v}_f p)^{\gamma-1}\Phi(1, 1+\gamma, -\frac{pr}{2})\right). \quad (52)
$$

Because of the Heaviside function, if $p \neq 0$, then $\mathrm{Im}G_+^R(\omega = 0) = 0$.

From the Kramers-Kronig relation, the real part of the Green function is given by $\mathrm{Re}G^R(\omega) = P\int_{-\infty}^{\infty} dz\frac{A(z)}{\omega-z}$ where $P$ denotes the Cauchy principal integral. We substitute in $A_+(p, \omega)$ from Eq. 21. The result is

$$
\mathrm{Re}G_+^R(\omega = 0) = -\frac{1}{\gamma\Gamma^2(\gamma)}\left(\frac{r}{2\tilde{v}_f}\right)^{2\gamma}
$$
$$
\times\int_{\tilde{v}_f|p|}^{\infty} dz\frac{e^{-\frac{zr}{\tilde{v}_f}}}{z}\left((z+\tilde{v}_f p)^\gamma(z-\tilde{v}_f p)^{\gamma-1}\Phi(1, 1+\gamma, \frac{r}{2\tilde{v}_f}(z+\tilde{v}_f p))\right.
$$
$$
\left. -(z-\tilde{v}_f p)^\gamma(z+\tilde{v}_f p)^{\gamma-1}\Phi(1, 1+\gamma, \frac{r}{2\tilde{v}_f}(z-\tilde{v}_f p))\right). \quad (53)
$$

The ratio between the first term and the second term of the integrand, without the minus sign, is

$$
R(p, z) = \frac{(z+\tilde{v}_f p)\Phi(1, 1+\gamma, \frac{r}{2\tilde{v}_f}(z+\tilde{v}_f p))}{(z-\tilde{v}_f p)\Phi(1, 1+\gamma, \frac{r}{2\tilde{v}_f}(z-\tilde{v}_f p))}. \quad (54)
$$

We note that if $R(p, z) \gtrless 1$, $\mathrm{Re}G_+^R(\omega = 0) \lessgtr 0$. One can determine the condition for which $R(p, z)$ is greater or less than 1 by using the power series expansion of the confluent hypergeometric function [28, 29],

$$
\Phi(\alpha, \beta, z) = \sum_{k=0}^{\infty}\frac{(\alpha)_k}{(\beta)_k}\frac{z^k}{k!}, \quad (55)
$$

where $(\lambda)_0 = 1$, $(\lambda)_k = \frac{\Gamma(\lambda+k)}{\Gamma(\lambda)}$, and $\beta$ cannot be a non-positive integer. Applying Eq. 55 to $\Phi(1, 1+\gamma, x)$, one has

$$
\Phi(1, 1+\gamma, x) = \sum_{k=0}^{\infty}\frac{\Gamma(1+\gamma)}{\Gamma(1+\gamma+k)}x^k. \quad (56)
$$

The coefficient $\frac{\Gamma(1+\gamma)}{\Gamma(1+\gamma+k)}$ is always positive if $\gamma > -1$. Therefore, $x\Phi(1, 1+\gamma, x)$ is an increasing function in $x$ for positive $x$. From the limit of integration in Eq. 53, we know that

$z \pm \tilde{v}_f p > 0$. We then need to compare $z + \tilde{v}_f p$ and $z - \tilde{v}_f p$. When $p > 0$, it is obvious that $z + \tilde{v}_f p > z - \tilde{v}_f p$. As a result, the numerator of $R(p,z)$ is greater than the denominator of $R(p,z)$ because $x\Phi(1, 1+\gamma, x)$ is an increasing function. In other words, $R(p,z) > 1$ when $p > 0$. Alternatively, when $p < 0$, one has $z + \tilde{v}_f p < z - \tilde{v}_f p$ and $R(p,z) < 1$ by the same reason. Consequently, the real part of the Green function changes sign at $p = 0$, i.e.,

$$\mathrm{Re}G_+^R(\omega = 0) \gtrless 0, \quad p \lessgtr 0. \tag{57}$$

Combining this result with $\mathrm{Im}G_+^R(\omega = 0) = 0$, we have

$$\phi_R(0) = -\pi + \pi\theta(-p) = -\pi + \pi\theta(G_+^R(p, 0)).$$

Hence, from Eq. 5, Luttinger's theorem of the right-movers is

$$n_+ = \int \frac{dp}{2\pi}\theta(-p) = \int \frac{dp}{2\pi}\theta(G_+^R(p, \omega = 0)). \tag{58}$$

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
