# Peer review of "Absence of Luttinger's theorem for fermions with power-law Green functions"

_SciPost Physics, doi:SciPost Phys. 5, 049 (2018)_

## Round 2 · Referee Report · Anonymous · 2018-4-12

Strengths

1) Addresses a question of importance to the frontier of the field
2) Solid analytic calculation
3) Well organized

Weaknesses

1) Concludes with general statements based on too specific special cases
2) Abstract and Introduction are somewhat overselling

Report

In this manuscript, the authors examine the validity of Luttinger’s theorem in strongly correlated systems, where the Fermi liquid paradigm (used in the original proof) breaks down. They study in detail two examples for systems of interacting Fermions, where the concept of Fermi sphere (and hence “Fermi sphere volume”) is not well defined. In the first, the Fermions Green’s function assumes an anomalous power-law dependence on frequency, where the deviation from Fermi-liquid is parametrized by an anomalous exponent \alpha, with \alpha->1 corresponding to the free Fermions limit. The second example is a 1D model, i.e. a Luttinger liquid. Based on direct analytic calculations of the Fermions density in these two cases, they conclude that Luttinger’s theorem does not hold in general but is recovered under certain restrictions. In particular, it does hold in the Luttinger liquid case (even when modified to break particle-hole symmetry), but is typically violated in the first example (although can be restored for 1<\alpha<2 by fine-tuning the low-energy cutoff). The breakdown of the theorem in the latter case is attributed to violation of a couple of criteria (particle-hole symmetry and vanishing of Im{G(\omega=0, -infinity}), which the authors conclude are necessary conditions for Luttinger’s theorem to hold.

In view of the growing interest in the behavior of electronic systems in the regime of no well-defined quasi-particles, the paper deals with an important question and is certainly relevant to this frontier. As far as I checked, the calculations are valid and I agree with the main results. However, I have several comments that the authors should address:
1) My primary concern is that the general conclusions drawn from the presented study, namely, the criteria stated as sufficient for Luttinger’s theorem to hold, are not fully substantiated. The manuscript presents a detailed derivation for two concrete cases, choosing specific functional forms of the Fermions Green’s function, and demonstrates that they are consistent with the stated criteria. However, to conclude that the criteria are sufficient, a direct proof is desired that an arbitrary G(p,\omega) satisfying them obeys Luttinger’s theorem. Unless the authors can come up with such direct proof, I believe their conclusions should be posed as a (plausible) conjecture.
2) The specific form of G(p,\omega) Eq. (3) is motivated by physical cases where the self-energy has anomalous \omega-dependence. However, it appears to be a somewhat artificial, relatively simple generalization of the free-Fermion Green’s function, which indeed enables an exact analytic calculation, but is not obviously compatible with any physical example. In particular, assuming a shift of the energy \epsilon_p by some self-energy of general form (e.g. an anomalous power of \omega) may, at best, reduce to Eq. (3) in some restricted regime of p,\omega, not the entire range required to compute the density n. Can the author address such an alternative form for the non-Fermi-liquid Green’s function?
3) As a minor remark, the abstract (4th line) states: “This contrasts with the result by Ref. [1,2]…”. I find it a bit misleading, since it creates the impression that the present paper contradicts the results of Ref. [1,2] while it actually agrees with them: violation of Luttinger’s theorem is shown for the power-law Eq. (3), not for a Luttinger liquid.

In summary, the paper is worth publishing in SciPost provided the authors address the above comments, and possibly modify some misleading statements.

Requested changes

1) Soften the general applicability of the conclusions (or provide a more convincing argument in their favor).
2) (Optional) If possible, address a more physical example of non-Fermi-liquid
3) Modify a misleading sentence in the Abstract as detailed in the report

---

## Round 4 · Author Response

We would like to resubmit ``Absence of Luttinger's theorem for fermions with power-law Green functions" which we have revised based on the referee's comments and suggestions. We thank the referees for carefully reviewing the manuscript. We start by responding to the the first comment.

  1. We agree that a rigorous proof is necessary to establish that the two properties (the vanishing of $\mathrm{Im}G(\omega)$ at $\omega = 0,-\infty$ and particle-hole symmetry) are necessary but not sufficient conditions for the Luttinger sum rule to hold. We now put this statement as a conjecture. We also softened the conclusion regarding the criteria for the validity of Luttinger's theorem.

  2. We added a new appendix (now Appendix B) to address a more physical example of non-Fermi liquid. We examined the validity of Luttinger's theorem of the system with self-energy of the form, $\Sigma \sim \lambda (\omega - \varepsilon_p)^\alpha$. We found that the Luttinger's theorem doesn't hold in general like the result we obtained with the power-law Green function.

  3. We modified this sentence in the abstract to ``However, Ref. [1,2] showed that Luttinger liquids, another scale-invariant fermionic model, respect Luttinger's theorem."

---

## Round 4 · List of Changes

1. We modified the sentence “This contrasts with the result by Ref. [1,2]…” to "However, Ref. [1, 2] showed that Luttinger liquids, another scale-invariant fermionic model, respect Luttinger’s theorem." in abstract.

2. We put the two properties (the vanishing of $\mathrm{Im}G(\omega)$ at $\omega = 0,-\infty$ and particle-hole symmetry) are sufficient but not necessary conditions as a conjecture in abstract, introduction, and conclusion.

3. We softened the conclusion regarding the applicability of the result.

4. In section II.D and (now) appendix B, we addressed a more physical example of non-Fermi liquid.

You are currently on this page

Resubmission 1708.08460v4 on 17 July 2018

---

## Editorial Decision

published